# Thermal Analysis on Active Heat Dissipation Design with Embedded Flow Channels for Flexible Electronic Devices

**DOI:** 10.3390/mi12101165

**Published:** 2021-09-28

**Authors:** Yanan Yu, Yafei Yin, Yuhang Li, Min Li, Jizhou Song

**Affiliations:** 1Institute of Solid Mechanics, Beihang University (BUAA), Beijing 100191, China; kiana_yuu@163.com (Y.Y.); 13231040@buaa.edu.cn (Y.Y.); limin@buaa.edu.cn (M.L.); 2Department of Engineering Mechanics, Soft Matter Research Center, and Key Laboratory of Soft Machines and Smart Devices of Zhejiang Province, Zhejiang University, Hangzhou 310027, China; 3State Key Laboratory of Fluid Power and Mechatronic Systems, Zhejiang University, Hangzhou 310027, China

**Keywords:** thermal analysis, flow channel, active heat dissipation, flexible electronics

## Abstract

Heat generation is a major issue in all electronics, as heat reduces product life, reliability, and performance, especially in flexible electronics with low thermal-conductivity polymeric substrates. In this sense, the active heat dissipation design with flow channels holds great promise. Here, a theoretical model, validated by finite element analysis and experiments, based on the method of the separation of variables, is developed to study the thermal behavior of the active heat dissipation design with an embedded flow channel. The influences of temperature and flow velocity of the fluid on heat dissipation performance were systematically investigated. The influence of channel spacing on heat dissipation performance was also studied by finite element analysis. The study shows that performance can be improved by decreasing the fluid temperature or increasing the flow velocity and channel density. These results can help guide the design of active heat dissipation with embedded flow channels to reduce adverse effects due to excessive heating, thus enhancing the performance and longevity of electronic products.

## 1. Introduction

In recent years, electronic products have emphasized intelligence and miniaturization, which require higher performance at the integration level. In particular, flexible electronic devices represent a rapidly developing branch, which has enabled many new applications such as curvilinear electronics and bio-integrated electronics [1,2,3,4,5,6] due to their unique advantages of stretchability and flexibility [7,8,9,10]. Furthermore, hybrid flexible systems have been investigated, such as electrophysiology [11], multimodal [12], large-area pressure [13] and other functions [14]; such functions may lead to overheating in the system. Heat generation is a major issue when these electronic devices are in service. According to the Arrhenius relation, the thermal failure rate of flexible electronic devices has an exponential relationship with temperature [15], in which the failure probability doubles for every 10 °C increase in temperature. Additionally, the use of polymer materials with poor thermal conductivity as the substrate creates more challenges regarding heat dissipation in flexible electronic devices.

In order to solve this special challenge, many thermal designs have been reported to improve heat dissipation in flexible electronic devices. Our group focused on passive thermal designs to control and tune the thermal flow in order to achieve adequate heat dissipation. Li et al. [16] reported on an orthotropic substrate design to control the heat flow paths. Shi et al. [17] proposed a phase change material enabled functional soft composite to serve as the thermal protection substrate for wearable electronics. Other groups have developed various new materials to enhance thermal conductivity. Jeong et al. [18] and Bartlett et al. [19] embedded liquid metal inclusions into a polymer to achieve both flexibility and high thermal conductivity. Sun et al. [20] designed a flexible graphene aerogel-based composite phase change film for personal thermal management applications. Although the above designs were all passive, they hold promise for the thermal management of flexible electronic devices. Compared to passive thermal designs, active designs, which have been extensively studied for silicon electronics, could be more efficient in heat dissipation [21,22,23], and are attracting more and more attention. At present, a substrate with embedded flow channels is the most common active heat dissipation design, where fluid flowing in the channel takes away heat. The earliest microchannel for water on a silicon plate, designed by Tuckeman and Pease in the 1980s, was able to achieve 800 W heat dissipation on a 1 cm^2^ chip [24]. Perret et al. [25] performed a numerical analysis of heat transfer in rectangular, rhombic and hexagonal flow channels and found that the former had the lowest thermal resistance. Yong et al. [26] added fins to the traditional rectangular channel and found that a smaller fin angle and smaller fin pitch could better improve heat dissipation performance. Husain and Kim [27] studied the influence of the width-depth ratio of the channel on the heat transfer of the system.

Due to the very poor thermal conductivity of polymer substrates, passive heat dissipation methods cannot provide adequate efficiency. Introducing active fluid flow in channels embedded in soft polymer substrates may significantly improve the efficiency of heat dissipation. However, few research efforts have sought to quantitatively investigate the effects of fluid flowing in the channels embedded in a soft polymer substrate. Because of the complexity and difficulty of solving the active heat dissipation problem of fluid–solid coupling, all the studies mentioned above were primarily numerical analyses, lacking theoretical thermal analyses of the structure with embedded flow channels.

This paper aims to develop a theoretical model to investigate the thermal behavior of an active heat dissipation design with embedded flow channels. Finite element analysis and experiments are also carried out to validate the analytical model. The paper is structured as follows. Section 2 describes the theoretical model. Section 3 presents the results and a discussion. Section 4 presents the conclusion.

## 2. Theoretical Model

The active heat dissipation design with flow channels can be applied to the substrate for flexible electronic devices. Figure 1a schematically shows a typical soft substrate, which is made of polydimethylsiloxane (PDMS), with an embedded flow channel for active heat dissipation. Since the diameter of the flow channel is usually much smaller than the characteristic length of the substrate (e.g., thickness), axisymmetric treatment of the problem is reasonable. The hollow channel is located at the center of the substrate. The fluid can flow in the channel along the direction indicated by the arrow changing from blue to red. Among the many liquid materials flowing in the channel, water is the most common, due to its nontoxicity, large specific heat capacity and availability. Figure 1b is a schematic diagram of the theoretical model. The origin of coordinates (*r*, *z*) is set at the center of the middle surface of the flow channel. *R*_c_ and *R*_s_ denote the channel radius and substrate radius, respectively. *H* is the length of the flow channel. Path 1, Path 2 and Path 3 represent the inlet surface, the fluid–solid contact surface and the symmetric surface, respectively.

The steady temperature field *T*(*r*, *z*) of the PDMS substrate satisfies the following axisymmetric heat transfer equation
(1)$$\frac{\partial^{2}T}{\partial r^{2}}+\frac{1}{r}\frac{\partial T}{\partial r}+\frac{\partial^{2}T}{\partial z^{2}}=0,0<z<\frac{H}{2},R_{c}<r<R_{s}$$

The fluid–solid coupling at the interface (*r* = *R*_c_) can be simplified as a forced convention, while the coefficient *h*_f_ of heat convention can be obtained by the Sieder-Tate formula as [28],
(2)$$h_{f}=\frac{k_{f}}{2R_{c}}1.86{\left(Re\cdot Pr\cdot \frac{2R_{c}}{H}\right)}^{1/3}$$
where $Re=\frac{\rho_{f}v\cdot 2R_{c}}{\mu_{f}}$ and $Pr=\frac{c_{f}\mu_{f}}{k_{f}}$ are the Reynolds number representing the convective strength and the Prandtl number representing relationship between temperature and flow boundary layers of the fluid, respectively, and *k*_f_, *ρ*_f_, *μ*_f_, *c*_f_ are the thermal conductivity, density, viscosity and specific heat capacity of the fluid, respectively. Since the radius of the flow channel is smaller than that of substrate, the fluid has a negligible influence on the outer boundary of the PDMS. Therefore, the boundary conditions of the PDMS along the radial direction can be expressed as
(3)$$\left\{ \begin{array}{l} {-k\frac{\partial T}{\partial r}|}_{r=R_{c}}=-h_{f}\left({T|}_{r=R_{c}}-T_{f}\right)\\ {T|}_{r=R_{s}}=T_{0} \end{array}\right.$$
where *k* and *T*_0_ are the thermal conductivity and initial temperature of the substrate, respectively, and *T*_f_ is the temperature of fluid.

The fluid temperature in the channel remains almost the same [29], such that the plane at *z* = 0 can be considered a symmetric plane. Moreover, a natural convection occurs at the upper surface of the PDMS. Thus, the boundary conditions along the axial direction can be expressed as:(4)$$\left\{ \begin{array}{l} {-k\frac{\partial T}{\partial z}|}_{z=0}=0\\ {-k\frac{\partial T}{\partial z}|}_{z=\frac{H}{2}}=h_{a}\left({T|}_{z=\frac{H}{2}}-T_{a}\right) \end{array}\right.$$
where *h*_a_ is the natural convective coefficient and *T*_a_ is the ambient temperature.

We define the temperature rise from ambient temperature as $\theta =T-T_{a}$. The heat transfer Equation (1) and boundary conditions in Equations (3) and (4) then become
(5)$$\frac{\partial^{2}\theta }{\partial r^{2}}+\frac{1}{r}\frac{\partial \theta }{\partial r}+\frac{\partial^{2}\theta }{\partial z^{2}}=0,\;0<z<\frac{H}{2},R_{c}<r<R_{s}$$
(6)$$\left\{ \begin{array}{l} {-k\frac{\partial \theta }{\partial r}|}_{r=R_{c}}=-h_{f}\left({\theta |}_{r=R_{c}}+T_{a}-T_{f}\right)\\ {\theta |}_{r=R_{s}}=T_{0}-T_{a} \end{array}\right.$$
(7)$$\left\{ \begin{array}{l} {-k\frac{\partial \theta }{\partial z}|}_{z=0}=0\\ {-k\frac{\partial \theta }{\partial z}|}_{z=\frac{H}{2}}=h_{a}{\theta |}_{z=\frac{H}{2}} \end{array}\right.$$

The method of separation of variables was adopted to solve the above heat transfer problem. The temperature rise *θ* takes the following form:(8)θ(r,z)=X(r)Y(z)
where *Y*(*z*) and *X*(*r*) satisfy
(9)$$\frac{\partial^{2}Y}{\partial z^{2}}+\gamma^{2}Y=0$$
and
(10)$$\frac{\partial^{2}X}{\partial r^{2}}+\frac{1}{r}\frac{\partial X}{\partial r}-\gamma^{2}X=0$$
with *γ* as the eigenvalue. Equations (7), (9) and (10) yield the solution of *θ* as:(11)$$\theta \left(r,z\right)=\sum_{n=1}^{\infty }\left[A_{n}I_{0}\left(\gamma_{n}r\right)+B_{n}K_{0}\left(\gamma_{n}r\right)\right]\cos\left(\gamma_{n}z\right)$$
where *I*_0_ and *K*_0_ is the 0-th order modified Bessel function of the first and second kinds, respectively, *γ*_n_ satisfies $k\gamma \sin\left(\gamma H/2\right)-h_{a}\cos\left(\gamma H/2\right)=0$, and coefficients *A*_n_ and *B*_n_ satisfy
(12)$$\left[\begin{array}{cc} I_{1}\left(\gamma_{n}R_{c}\right)\gamma_{n}k-I_{0}\left(\gamma_{n}R_{c}\right)h_{l} & -\left[K_{1}\left(\gamma_{n}R_{c}\right)\gamma_{n}k+K_{0}\left(\gamma_{n}R_{c}\right)h_{f}\right]\\ I_{0}\left(\gamma_{n}R_{s}\right) & K_{0}\left(\gamma_{n}R\right) \end{array}\right]\left\{\begin{array}{c} A_{n}\\ B_{n} \end{array}\right\}=\left\{\begin{array}{c} h_{f}\left(T_{a}-T_{f}\right)\\ T_{0}-T_{a} \end{array}\right\}\frac{G_{n}}{N_{n}}$$

Here, $N_{n}=\int_{z=0}^{H/2}Y^{2}\left(\gamma_{n},z\right)\mathrm{dz},G_{n}=\int_{z=0}^{H/2}Y\left(\gamma_{n},z\right)\mathrm{dz}$, *I*_1_ and *K*_1_ are the first-order modified Bessel function of the first and second kinds, respectively. Coefficients *A*_n_ and *B*_n_ can be given by:(13)$$\{\begin{array}{c} A_{n}=\frac{G_{n}\left[K_{1}\left(\gamma_{n}R_{c}\right)\gamma k\left(T_{0}-T_{a}\right)+K_{0}\left(\gamma_{n}R_{c}\right)h_{f}\left(T_{0}-T_{a}\right)+K_{0}\left(\gamma_{n}R_{s}\right)h_{f}\left(T_{a}-T_{f}\right)\right]}{N_{n}\left[\begin{array}{l} \gamma_{n}k\left(K_{0}\left(\gamma_{n}R_{s}\right)I_{1}\left(\gamma_{n}R_{c}\right)+I_{0}\left(\gamma_{n}R_{s}\right)K_{1}\left(\gamma_{n}R_{c}\right)\right)\\ -h_{f}\left(K_{0}\left(\gamma_{n}R_{s}\right)I_{0}\left(\gamma_{n}R_{c}\right)-I_{0}\left(\gamma_{n}R_{s}\right)K_{0}\left(\gamma_{n}R_{c}\right)\right) \end{array}\right]}\\ B_{n}=\frac{G_{n}\left[I_{1}\left(\gamma_{n}R_{c}\right)\gamma_{n}k\left(T_{0}-T_{a}\right)-I_{0}\left(\gamma_{n}R_{c}\right)h_{f}\left(T_{0}-T_{a}\right)-I_{0}\left(\gamma_{n}R_{s}\right)h_{f}\left(T_{a}-T_{f}\right)\right]}{N_{n}\left[\begin{array}{l} \gamma_{n}k\left(K_{0}\left(\gamma_{n}R_{s}\right)I_{1}\left(\gamma_{n}R_{c}\right)+I_{0}\left(\gamma_{n}R_{s}\right)K_{1}\left(\gamma_{n}R_{c}\right)\right)\\ -h_{f}\left(K_{0}\left(\gamma_{n}R_{s}\right)I_{0}\left(\gamma_{n}R_{c}\right)-I_{0}\left(\gamma_{n}R_{s}\right)K_{0}\left(\gamma_{n}R_{c}\right)\right) \end{array}\right]} \end{array}$$

The temperature field T(r,z) can then be obtained as:(14)$$T\left(r,z\right)=T_{a}+\theta \left(r,z\right)=T_{a}+\sum_{n=1}^{\infty }X_{n}\left(r\right)Y_{n}\left(z\right)$$

## 3. Results and Discussion

### 3.1. Model Verification

We carried out experiments and finite element analysis (FEA) to validate our theoretical model. Figure 2a shows the experimental setup. A flow channel with a radius *R*_c_ of 1 mm and a length *H* of 60 mm was embedded in a PDMS substrate with the radius *R*_s_ of 25 mm. The inlet was connected to an injector containing ice in order to keep the fluid cold. The cold fluid was pumped to the flow channel by an injection pump at a flow rate of 90 mL/min, and then flowed out through the PDMS into a liquid storage tank. The temperature of the PDMS was measured using an infrared thermal imager. Figure 2b shows the infrared thermal imager measurements at the inlet of channel. The fluid temperature at the inlet was 4.3 °C and the environmental temperature *T*_a_ was 24.6 °C. Figure 2c shows the temperature distribution on the upper inlet surface of the structure.

The fluid–solid coupling FEA was performed using the commercial software, FLUENT. The thermal conductivity *k* of PDMS was 0.17 W/(K·m). The parameters of fluid (i.e., water) were set to 0.6 W/(K·m) for the thermal conductivity *k*_f_, 998.2 Kg/m^3^ for the mass density *ρ*_f_, 0.001003 Pa·s for the viscosity *μ*_f_, and 4182 J/(kg·K) for the specific heat capacity c_f_. The initial temperature *T*_0_ was equal to the ambient temperature *T*_a_; both were set to 24.6 °C. The coefficient of thermal convection *h*_a_ was defined as 20 W/(K·m^2^) [30].

In order to measure the heat dissipation capacity of the flow channel, we defined a relative temperature drop affected by flow channel as:(15)$$Q=\frac{T_{0}-T}{T_{0}-T_{f}}\times 100\%$$

Figure 3 shows the temperature along Path 1, where the bar representing the experimental error was obtained from three experiments. The theoretical predictions agreed well with the FEA and experiments, which validates the accuracy of our theoretical model. As expected, the temperature increased as the distance to the channel increased, which yielded a decreasing heat dissipation capacity. For example, at *r* = 2 mm, the temperature of PDMS was 12.3 °C, and Q=61.3%. When *r* increased to 6 mm, the temperature increased to 19.7 °C and *Q* decreased to Q=25.6%. These results indicate that the flow channel significantly reduced the temperature of the PDMS. If a relative temperature drop of 25% was set as a criterion for heat dissipation, the influence range of flow channel on temperature was six times its own radius *R*_c_.

### 3.2. Influences of Fluid Temperature and Flow Velocity

In order to investigate the influence of fluid temperature on the thermal behavior of the active heat dissipation design, theoretical predictions and FEA were carried out with *T*_f_ of 5/10/15 °C. The initial and ambient temperature were 25 °C and the fluid velocity was 100 mm/s. Figure 4a shows the temperature along Path 2. Again, there was good agreement between the theoretical prediction and FEA and the maximum temperature difference of 0.56 °C. As expected from FEA, the temperature of the PDMS decreased slightly along the flowing direction, since the heat was taken away by the flowing fluid. This slight temperature change validated the theoretical treatment of the influence of flowing fluid as a forced convection with a constant convective coefficient, although the local convective coefficient decreased along the flowing direction. Figure 4b compares the temperature along Path 3 according to the theoretical prediction and FEA with various fluid temperatures of 5/10/15 °C. It can be observed that the theoretical prediction agreed well with FEA. As expected, lower fluid temperature yielded better heat dissipation performance. At the location of *r* = 10 mm, the temperatures of PDMS were 19.6 °C, 21 °C and 22.3 °C for fluid temperatures of 5 °C, 10 °C and 15 °C respectively. It was interesting to find that although the temperatures at the same location were different for different fluid temperatures, the temperature drop percentages *Q* were all 27%, which indicated that the influence range of flow channel on heat dissipation was not affected by the fluid temperature *T*_f_.

Figure 5a shows the convective coefficient on the flow channel wall as a function of flow velocity from the Sieder-Tate formula. As shown, the convective coefficient increased with increasing fluid velocity. When the flow velocity increased from 20 mm/s to 800 mm/s, the coefficient *h*_f_ increased from 1172 W/(K·m^2^) to 4010 W/(K·m^2^). In addition, the convective coefficient *h*_f_ increased rapidly when flow velocity *v* was small, while the changing of coefficient *h*_f_ became slow when *Re* exceeded 640. These results indicated that the flow velocity may affect the heat dissipation performance.

To demonstrate the influence of flow velocity on the thermal behavior of the active heat dissipation design, we show the temperature at the feature point, (*R*_c_, 0) as a function of flow velocity *v* in Figure 5b, where the fluid temperature *T*_f_ was 5/10/15 °C. It is shown that the FEA results were all higher than the theoretical predictions, since the actual higher local temperatures of the fluid at the fluid–solid interface were accounted for in FEA corresponding to a lower value of *T*_l_ at the inlet in the theoretical model. When *T*_f_ was 5 °C, there was a more distinct difference between the FEA and theoretical prediction. The temperature differences were 0.4 °C for the *Re* of 120 and 0.14 °C for 780. This temperature difference decreased with *Re* because the added heat in the fluid, when flowing from the inlet to the feature point (*R*_c_, 0), increased as the flow velocity *v* slowed, resulting in a higher fluid temperature relative to the *T*_f_. Considering that the temperature differences between FEA and theoretical predictions were all lower than 0.4 °C, it can be still considered that the theoretical predictions and FEA were in good agreement.

### 3.3. Thermal Analysis of Active Heat Dissipation Design with Multiple Flow Channels

The previous results and discussion were for the active heat dissipation design with a single flow channel. In order to further enhance the heat dissipation performance, multiple channels are usually adopted in practical applications. Figure 6a shows a cross-section of the active heat dissipation design with multiple flow channels, with *l* representing the spacing of neighboring channels. As an analysis of the modelling of multiple channels was not possible, only FEA was carried out here to investigate the thermal behavior. Due to the symmetric conditions, a single unit cell was developed with a flow channel (radius: 1 mm) located in the middle of a square PDMS (in-plane dimension: *l* × *l*). The length of the flow channel was 60 mm. The bottom and upper sides were set as natural convection boundaries with a coefficient of 20 W/(K·m^2^), and the remaining sides were set as the adiabatic boundaries. The initial and ambient temperatures were both set at 25 °C. The fluid temperature was 5 °C.

Figure 6b shows the temperature distribution on the cross-section of the unit cell with a spacing *l* of 30 mm. *s* is defined as the distance between the point on the diagonal and the center. The maximum temperature, which occurred at the four corners of the unit cell, was 11.2 °C, while the minimum temperature, which occurred at the fluid–solid interface, i.e., where the related temperature drop *Q* ranged from 69% to 96.5%, was 5.7 °C. Figure 6c shows the temperature distribution along the diagonal with spacing *l* in the range of 10 to 90 mm. The maximum temperature increased with the increase of spacing, since a larger spacing yields a lower heat dissipation capacity, resulting in a higher temperature. When the channel spacing *l* was 90 mm, the PDMS temperature was 19.6 °C, and *Q* was only 27% at the position of 15 times the channel radius away from the center compared to the same temperature at the position six times the channel radius for the case of a single flow channel. Thus, the heat dissipation influence of multiple channels is much better than that of a single channel. Figure 6d shows the maximum and minimum temperatures on the cross-section with respect to the channel spacing *l*. The channel spacing had a significant influence on the maximum temperature but a negligible influence on the minimum temperature. When the spacing increased from 10 mm to 90 mm, the maximum temperature increased by 350%, i.e., from 5.2 °C to 23.4 °C, and *Q* decreased from 99% to 8%, while the minimum temperature only increased by 18%, i.e., from 5.0 °C to 5.9 °C, and *Q* was almost constant. Moreover, when the spacing was larger than the influence range of a single channel, its effect became negligible. However, under the same flow rate, the smaller spacing meant that the number of channels increased, which made it necessary to improve the pump power to overcome the pressure drop. Therefore, in practical applications, the contradiction between the number of channels and cooling performance needs to be adjusted so that cooling capacity and energy saving are maximized.

## 4. Conclusions

In summary, a theoretical model based on the method of separation of variables was developed to investigate the thermal behavior of an active heat dissipation design with an embedded flow channel. The fluid–solid coupling at the interface was simplified, i.e., to forced convention. Finite element analysis and experiments were also carried out to validate the analytical model. It was found that the temperature and velocity of the fluid had distinct heat dissipation effects on the system. Specifically, decreasing the former and increasing the latter significantly improved the heat dissipation performance. Furthermore, the active heat dissipation design with multiple channels was also studied by finite element analysis. It was found that denser flow channels have a much better heat dissipation performance.

## Figures and Tables

**Figure 1 micromachines-12-01165-f001:**
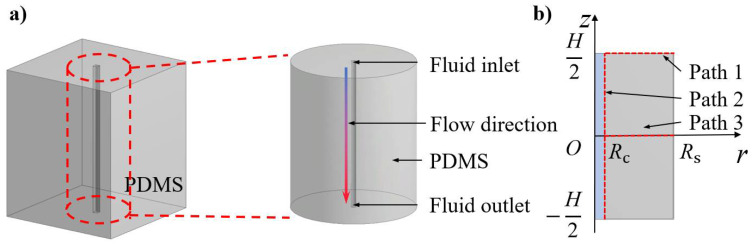
Schematic diagrams of (**a**) a typical active heat dissipation design with an embedded flow channel and the axisymmetric treatment of the problem, and (**b**) the theoretical model.

**Figure 2 micromachines-12-01165-f002:**
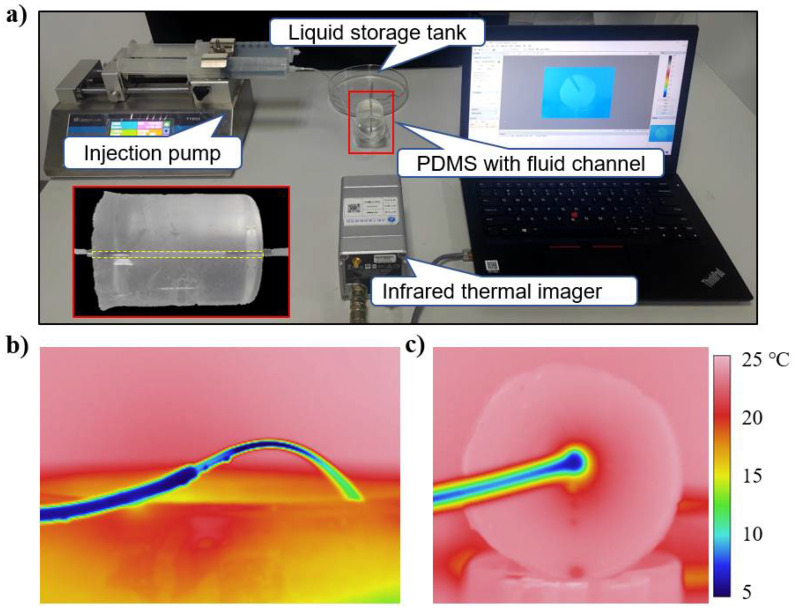
(**a**) Image of the experimental setup. (**b**) Thermal image at the inlet of channel. (**c**) Thermal image of the inlet surface.

**Figure 3 micromachines-12-01165-f003:**
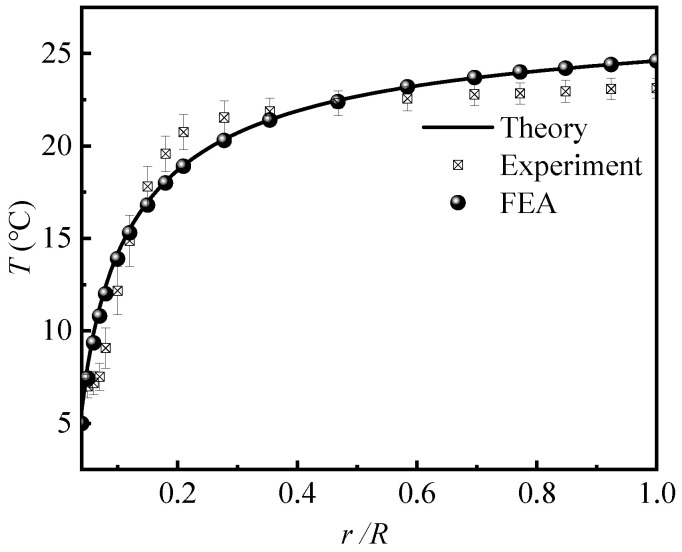
Temperature distribution along Path 1.

**Figure 4 micromachines-12-01165-f004:**
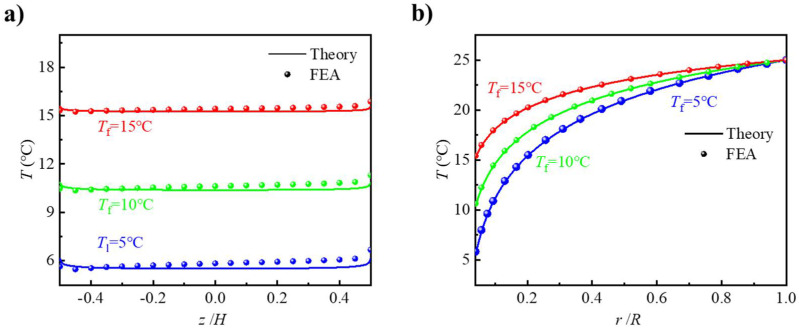
Temperature distributions along (**a**) Path 2 and (**b**) Path 3.

**Figure 5 micromachines-12-01165-f005:**
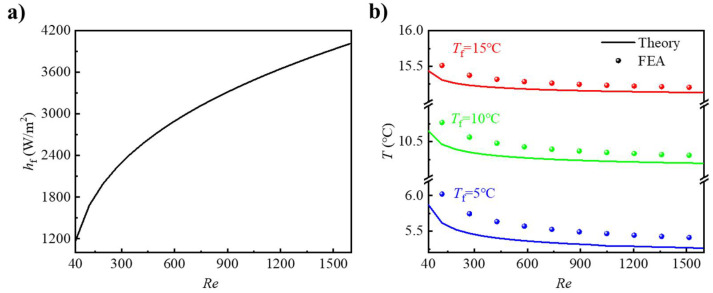
(**a**) Convective coefficient of the fluid–solid interface as a function of flow velocity. (**b**) The temperature at (*R*_c_, 0) as a function of flow velocity.

**Figure 6 micromachines-12-01165-f006:**
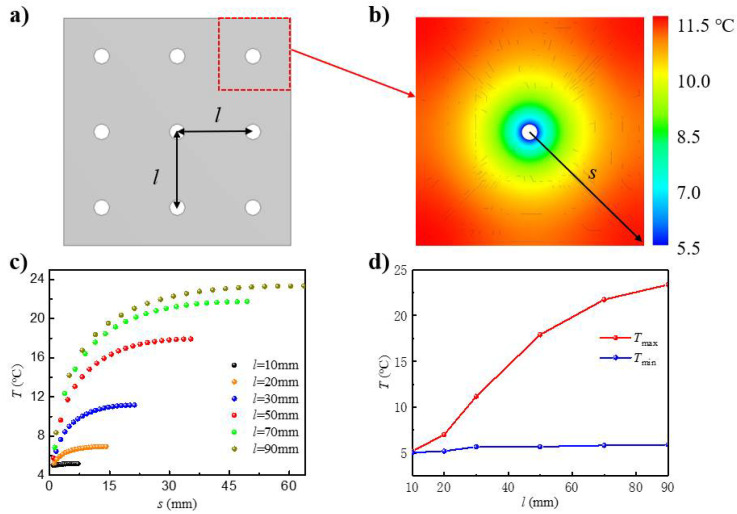
(**a**) Cross-sectional schematic of the active heat dissipation design with multiple flow channels in the soft substrate. (**b**) The temperature contour on the cross-section of the unit cell from FEA. (**c**) Temperature distribution along the diagonal with various channel spacings. (**d**) Maximum (red) and minimum (blue) temperatures as functions of the channel spacing.

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
