# Peer review of "Thermal Analysis on Active Heat Dissipation Design with Embedded Flow Channels for Flexible Electronic Devices"

_micromachines, 2021, doi:10.3390/mi12101165_

Round 1

Reviewer 1 Report

This manuscript studies the thermal analysis both in theoretical simulation and experiment of active heat dissipation design with embedded flow channels for flexible electronic devices. A network of flow channels of water in the PDMS matrix is used to investigate the heat dissipation with different environment temperatures and flow temperatures and speed. This manuscript is well prepared, results are clearly presented. The manuscript is possibly accepted for publication after a minor revision as follows:

  1. The novelty and necessity of this research are not well presented in the introduction.
  2. The results are not clearly mentioned in the abstract and conclusion.
  3. Why did the author use water for heat dissipation? please mention the advantages of water compared to other solvents.
  4. A comparison between the author's research and other previous research should be presented.
  5. Thermal image of the full length of the sample (digital photo in Fig 2a) should be provided.
  6. In reality, the temperature in the electronic devices or systems may be raised to 80 oC, an investigation of ambient temperatures as high as 25 oC is not enough. The author should investigate the effect of higher ambient temperature on heat dissipation by theoretical model and experiment.
  7. Figure 5 should be arranged in a horizontal direction.
  8.  The author should double-check the ref again. there are some mistyping that can be found (ref. 14, 16, 29, etc.)

Author Response

Response to Reviewer 1 Comments

Point 1: The novelty and necessity of this research are not well presented in the introduction.

 Response 1: We thank the review’s suggestion and added the sentences below in lines from 56 to 61.

Due to the very poor thermal conductivity in the polymer substrate, the passive heat dissipation methods cannot provide enough efficiency. Introducing active fluid flow-ing in the channels embedded into soft polymer substrate may help strongly improve the efficiency of the heat dissipation. However, there is very less research articles to quantitatively investigate the effects of fluid flowing in the channels embedded in the soft polymer substrate.

Point 2: The results are not clearly mentioned in the abstract and conclusion.

Response 2: Thank you for your constructive advice. The Eq. (6b) has been denoted by Eq. (5b), it is satisfied with the general contact. Please see the red sentences on page 8 in the revised version. Relevant results have been supplemented in the abstract and conclusion. The details are in lines 18 and 262.

 Point 3: Why did the author use water for heat dissipation? please mention the advantages of water compared to other solvents

Response 3: Thank you for your valuable advice. We have added the advantages of water, such as non-toxicity and low price in line 18. And this work focuses on the analytical thermal analysis, which does not consider the specific properties of the liquid and we may consider other solvents in the future.

Point 4: A comparison between the author's research and other previous research should be presented.

Response 4: Thank you for your constructive suggestion. We have added some comparisons from Line 40-46.

Point 5: Thermal image of the full length of the sample (digital photo in Fig 2a) should be provided.

Response 5: Thank you for your scientific advice. During the experiment, the thermal imager can only record the temperature results of one visual angle, so there is no sample for full length.

Point 6: In reality, the temperature in the electronic devices or systems may be raised to 80 ℃, an investigation of ambient temperatures as high as 25 ℃ is not enough. The author should investigate the effect of higher ambient temperature on heat dissipation by theoretical model and experiment.

Response 6: Thank you for your scientific advice. We also considered the influence of ambient temperature. But considering the stage of the study, this paper only considered the most basic situation, and the study of the influence of ambient temperature or other heating patch components will be arranged in the future.

Point 7: Figure 5 should be arranged in a horizontal direction.

Response 7: Thank you for your scientific advice. We have rearranged the Figure 5.

Point 8: The author should double-check the ref again. there are some mistyping that can be found (ref. 14, 16, 29, etc.)

Response 8: Thank you for your scientific advice. We have corrected them.

Reviewer 2 Report

The article “Thermal analysis on active heat dissipation design with embedded flow channels for flexible electronic devices” highlights analytical results, FEA simulation, and experimental measurements for active heat dissipation of flexible electronic materials (e.g., PDMS). Overall, this article has developed (1) an analytical solution based on joint conduction-convection heat transfer, (2) performed FEA simulation for single and multiple micro-channels, and (3) validated temperature profiles against real-world experimental measurements. The research work is of significant interest to the electronics cooling community and, therefore, suitable for publication in Micromachines (special issue: Heat Management in Microdevices). However, the following comments/concerns must be addressed before publishing the article:

  1. Please avoid lump references in the manuscript such as [1-14]. The references should be individually cited and reviewed.
  2. In the first sentence of the abstract, please consider writing: “ Heat generation is a major issue…..”
  3. Reference [15] is not suitable for the statement claiming, “According to the Arrhenius relation, the thermal failure rate of flexible electronic devices has an exponential relationship with temperature [15],” Insert a proper citation to support this statement.
  4. Presently, the Introduction section does not review the state-of-the-art literature properly in the realm of active heat dissipation for electronics cooling. The Introduction should clearly articulate (a) review of the recent works, (b) identify gaps, and (c) demonstrate the novelty of the current work. A concise literature review Table highlighting key features/advantages/disadvantages of prior works can better motivate the importance of the article. In my opinion, it should be much more detailed with a smoother transition across paragraphs.
  5. For the separation of variables (Eq. 8 and onwards), consider replacing ‘Z’ and ‘R’ with ‘X’ and ‘Y’. Since Z and R are already defined as the coordinate system, it is misleading.
  6. What do the error bars in Figure 3 signify? The experimental uncertainties should be clearly stated. How many measurements were performed for calculating the error bars? % of experimental uncertainties should be noted within the discussion.
  7. In Figures 3 and 4, the ‘r’ and ‘z’ axes should be represented dimensionless as r/R and z/H since those are known dimensions of cylinder radius and height.
  8. In Figure 5, the velocity axis should be represented in terms of the corresponding Reynolds numbers.
  9. It is instructive to convert the temperature axis of Figure 5b in terms of thermal resistance (K/W). This will help conclude: “Increasing the Reynolds number lowers the thermal resistance.”
  10. Express temperatures in Figures 3, 4, 5, and 6 in terms of dimensionless temperature: (T-T_a/(T_f-T_a)
  1. It seems that there is no mismatch between the Theoretical prediction and FEA in Figure 4b for various T_f with v=100 mm/s. But there is a mismatch between the temperature profiles shown in Figure 5. What is the reason?
  2. In section 3.3, it was inferred that increasing the number of channels will lower the temperature, which is better for the target heat-dissipating electronic device required to be cooled. However, it is well known that increasing the number of flow channels might paradoxically increase the pumping power to overcome the pressure drop. A discussion on this aspect is required at the end of section 3.3, highlighting heat transfer enhancement at the cost of pumping power.
  3. Careful proofreading must be done throughout the manuscript.

Author Response

Response to Reviewer 2 Comments

Point 1: Please consider clarifying/revising these in the texts as well as in the title. Please avoid lump references in the manuscript such as [1-14]. The references should be individually cited and reviewed.

 Response 1: Thank you very much for your kind suggestion. We have separated them in the revised manuscript.

Point 2: In the first sentence of the abstract, please consider writing: “ Heat generation is a major issue…..”

Response 2: Thank you for your scientific advice. We have replaced the original word with the suggested one in line 10.

 Point 3: Reference [15] is not suitable for the statement claiming, “According to the Arrhenius relation, the thermal failure rate of flexible electronic devices has an exponential relationship with temperature [15],” Insert a proper citation to support this statement.

Response 3: Thank you for your scientific advice. We have replaced the reference [15] with new one and the specific title is in line 306.

Point 4: Presently, the Introduction section does not review the state-of-the-art literature properly in the realm of active heat dissipation for electronics cooling. The Introduction should clearly articulate (a) review of the recent works, (b) identify gaps, and (c) demonstrate the novelty of the current work. A concise literature review Table highlighting key features/advantages/disadvantages of prior works can better motivate the importance of the article. In my opinion, it should be much more detailed with a smoother transition across paragraphs.

Response 4: Thank you for your scientific advice. We have revised the manuscript in the introduction.

Point 5: For the separation of variables (Eq. 8 and onwards), consider replacing ‘Z’ and ‘R’ with ‘X’ and ‘Y’. Since Z and R are already defined as the coordinate system, it is misleading.

Response 5: Thank you for your scientific advice. We have replaced the ‘Z’ and ‘R’ with ‘Y’ and ‘X’ in line 113.

Point 6: What do the error bars in Figure 3 signify? The experimental uncertainties should be clearly stated. How many measurements were performed for calculating the error bars? % of experimental uncertainties should be noted within the discussion.

Response 6: Thank you for your scientific advice. The error bars represent the experiments error from three experiments.

Point 7: In Figures 3 and 4, the ‘r’ and ‘z’ axes should be represented dimensionless as r/R and z/H since those are known dimensions of cylinder radius and height.

Response 7: Thank you for your scientific advice. We have rearranged the axes by r/R and z/H in figures 3 and 4.

Point 8: In Figure 5, the velocity axis should be represented in terms of the corresponding Reynolds numbers.

Response 8: Thank you for your scientific advice. We have replaced the velocity axis with Reynolds numbers axis in figure 5.

Point 9: It is instructive to convert the temperature axis of Figure 5b in terms of thermal resistance (K/W). This will help conclude: “Increasing the Reynolds number lowers the thermal resistance.”

Response 9: Thank you for your scientific advice. The theoretical thermal resistance can be calculated by theory and liquid convection coefficient, but the experimental one can not be obtained because of the lack of heat flow density.

Point 10: Express temperatures in Figures 3, 4, 5, and 6 in terms of dimensionless temperature: (T-T_a/(T_f-T_a)

Response 10: Thank you for your scientific advice. We have added the relevant dimensionless temperature statement in line 230.

Point 11: It seems that there is no mismatch between the Theoretical prediction and FEA in Figure 4b for various T_f with v=100 mm/s. But there is a mismatch between the temperature profiles shown in Figure 5. What is the reason?

Response 11: Thank you for your scientific advice. The ordinate ranges of the two figures are different. The data in Figure 5 are actually the first point in Figure 4. Therefore, there is indeed mismatch between theoretical results and FEA but not obvious under the ordinate range of Figure 5.

Point 12: In section 3.3, it was inferred that increasing the number of channels will lower the temperature, which is better for the target heat-dissipating electronic device required to be cooled. However, it is well known that increasing the number of flow channels might paradoxically increase the pumping power to overcome the pressure drop. A discussion on this aspect is required at the end of section 3.3, highlighting heat transfer enhancement at the cost of pumping power.

Response 12: Thank you for your scientific advice. We have simply supplemented the relationship between power and performance in line 245.

Point 13: Careful proofreading must be done throughout the manuscript.

Response 13: Thank you very much for your kind suggestion. We have checked the manuscript again.

Round 2

Reviewer 1 Report

The authors have solved problems that were raised. This manuscript is highly recommended to publish on Micromachines.